# Differential *ex vivo* susceptibility of *Plasmodium malariae* and *Plasmodium falciparum* clinical isolates from Ghana and Mali to current and lead discovery candidate antimalarial drugs

Alamissa Soulama,[1] Fanta Sogore,[2] Felix Ansah,[1] Ousmaila Diakite,[2] Jersley D. Chirawurah,[1] Fatoumata O. Maiga,[2] Mohamed Maiga,[2] Harry A. Danwonno,[1] Brice Campo,[3] Abdoulaye A. Djimde,[2] Gordon A. Awandare,[1] Lucas N. Amenga-Etego,[1] Laurent Dembele,[2] Yaw Aniweh[1]

**ABSTRACT** Non-falciparum species causing malaria in humans are considered neglected in the fight toward malaria elimination. Recent data highlight the increasing contribution of *Plasmodium malariae* to malaria morbidity and mortality. In this study, the susceptibility of *P. malariae* and *Plasmodium falciparum* to current antimalarial drugs was compared to advanced lead candidate drugs using field isolates. The blood samples were collected from the Central region of Ghana and Faladje and Kati in Mali. Following this, an *ex vivo* drug efficacy assay was conducted by screening mono-infected isolates against a panel of antimalarials. In Ghana, the susceptibility of the two species to most of the current antimalarial drugs was comparable, except for artemether, sulfadoxine, and atovaquone, for which the drugs were less potent against *P. malariae* than *P. falciparum* (7.12 vs 2.15 nM, 25.72 vs 7.86 nM, and 10.38 vs 2.51 nM, respectively). In Mali, quinine was significantly more potent against *P. malariae* than *P. falciparum* (18.35 and 26.84 nM), and tafenoquine was less potent against *P. malariae* than *P. falciparum* (5.50 and 2.85 nM). Among the candidate drugs, except INE963, whose inhibitory potency was comparable between both species, the other compounds significantly inhibited *P. malariae* more than *P. falciparum*. The data showed that current drugs investigated against the isolates from Ghana may be suitable for curing *P. malariae* infections. However, in Mali, chloroquine resistance appeared to have affected the suitability of quinine-based compounds for non-falciparum malaria treatment. Therefore, additional studies are required to establish the efficacy of artemether-lumefantrine for the treatment of *P. malariae* infections.

**IMPORTANCE** One major hurdle to research in the community is our inability to have continuous culture for parasites such as *Plasmodium malariae* and *Plasmodium ovale*. These two are common in the West African region and co-occur with *Plasmodium falciparum* in driving both clinical or asymptomatic infections as either mono-infections or mixed infections. This manuscript is a buildup of our efforts at using *ex vivo* methods to study the susceptibility of *P. malariae* and *P. falciparum* to conventional and lead compounds, comparing the isolates from Ghana and Mali. This is necessary to facilitate drug discovery efforts in combating malaria holistically. The community will greatly see this work as a step in the right direction, stimulating more research into these other parasites causing malaria.

**KEYWORDS** antimalarial agents, malaria, *Plasmodium*, non-falciparum

Malaria continues to be a life-threatening disease in sub-tropical regions, particularly in sub-Saharan Africa. Globally, all the control strategies against malaria have

**Peer Reviewer** Diogo R. M. Moreira, Instituto Goncalo Moniz, Salvador, Bahia, Brazil

Address correspondence to Laurent Dembele, laurent@icermali.org, or Yaw Aniweh, yaniweh@ug.edu.gh.

Alamissa Soulama and Fanta Sogore contributed equally to this article. Author order was determined by alphabetical order.

The authors declare no conflict of interest.

aimed to prevent mortality and morbidity and ultimately to reach disease elimination in endemic areas. Between 2000 and 2022, 2.1 billion cases and 11.7 million deaths have been averted, mainly due to the implementation of deliberate, well-coordinated, and integrated strategies (World Health Organization, 2023). The use of drugs has been very pivotal in the prevention, as well as in the case management strategies for malaria. Hence, about 4 billion treatment courses of Artemisinin Combination Therapy (ACT) have been delivered by the manufacturers between 2010 and 2022 (World Health Organization, 2023). However, the impact of the ACTs is being hampered by the emergence of drug-resistant *Plasmodium falciparum* parasites across malaria-endemic regions. Currently, partial artemisinin resistance has been reported in several African countries (1).

Human malaria can be caused by *P. falciparum*, *P. vivax*, *P. malariae*, *P. ovale curtisi*, *P. ovale wallikeri*, and *P. knowlesi*. In addition to these, there have been reports of *P. cynomolgi* infecting humans (2–4). The dominant *Plasmodium* species that cause malaria in the sub-Saharan region are mainly *P. falciparum*, *P. malariae*, *P. ovale curtisi*, *P. ovale wallikeri*, and *P. vivax* (5–9). However, *P. falciparum* accounts for the high disease burden, compared to the other species termed non-falciparum, whose contribution to the disease burden is often poorly studied. The need to evaluate the true contributions of these non-falciparum species to the burden of malaria is important because *P. malariae* is the second dominant parasite causing malaria in Mali and Ghana. Therefore, these non-falciparum species can pose a threat to the current malaria treatment strategies in malaria-endemic regions that mainly focus on *P. falciparum* infections. Recent reports show an increased frequency and prevalence of *P. malariae* in malaria-endemic areas such as the Amazon and sub-Saharan Africa (6, 10–13). Even though *P. malariae* infection results in benign disease, the long-term effects of *P. malariae* infection cause complications such as splenomegaly (14) or kidney damage (15), leading to hospitalization and death (9, 16). The current recommendation of the World Health Organization (WHO) for the treatment of *P. malariae* malaria is based on the use of ACTs or chloroquine, as no evidence of resistance to these drugs has been reported for *P. malariae*. However, studies have reported the persistence of *P. malariae* infections after ACT treatment (17, 18). Our recently published *ex vivo* study showed a reduced susceptibility of *P. malariae* to artemisinin and lumefantrine (6). There is a lack of sufficient data regarding the efficacy of the current antimalarials for the treatment of *P. malariae* infections.

In accordance with the WHO recommendations on *in vitro/ex vivo* monitoring of the efficacy of antimalarial drugs, this study was carried out to evaluate the susceptibility of *P. malariae*, in comparison with *P. falciparum*, to current antimalarial drugs against clinical isolates from Ghana and Mali. The antimalarials used in this study include artemether, lumefantrine, quinine, sulfadoxine, pyrimethamine, atovaquone, chloroquine, and tafenoquine. In addition, growth inhibitory assays were used to assess the potency of piperaquine and four promising advanced antimalarial lead candidate drugs against clinical isolates from Mali. Among these candidate drugs is INE963, an imidazothiadiazole derivative, which exhibited high potential in preclinical development and reached the first stage of clinical trials (19), while the three other compounds involved were MMV products that showed interesting results in preclinical stages (MMV1579167, MMV158179, and MMV1793609) (20).

## MATERIALS AND METHODS

### Study design

The samples from Ghana were collected as part of a cross-sectional study that is exploring antimalarial effectiveness on case management. As shown in Fig. 1, some of the samples were obtained from study participants who were screened for malaria in the community (Ewim). Briefly, volunteers were invited to be tested for malaria using rapid diagnosis test (RDT) and microscopy. After obtaining information about their socio-demographic profile and symptoms, blood samples were collected from

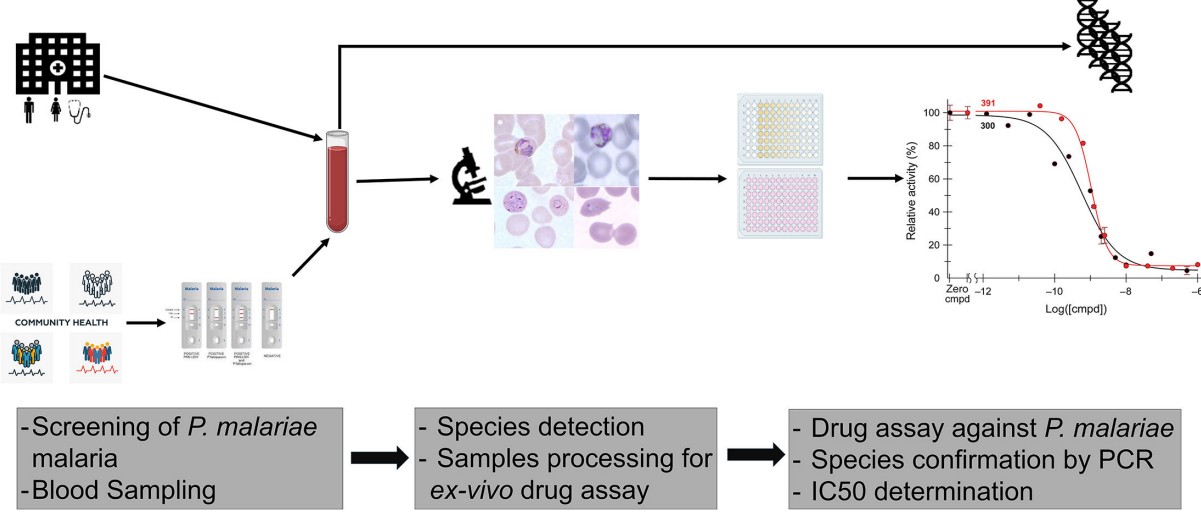

**FIG 1** Scheme of the study. The figure shows the processes that the samples are taken through from their collection to conducting the drug assay and representation of the data.

eligible volunteers who consented to participate in the study. The rest of the samples were collected from participants who were attending the outpatient department of the hospital (Ewim Polyclinic). The study participants were recruited based on symptomatic malaria, confirmed by RDT and microscopy. A volume of 5–10 mL of venous blood was collected from each volunteer who consented to be part of the study. In Mali, the samples for the assay were obtained from a cross-sectional screening and detection of all *Plasmodium* malaria cases during a longitudinal prospective study. This study aimed to assess the *ex vivo* efficacy of a panel of antimalarials against *P. malariae*. Thus, only freshly collected field isolates of *P. malariae* mono-infection parasites were used in the current study. The polymerase chain reaction (PCR) assay was used to detect and distinguish *P. malariae* mono-infection samples from other *Plasmodium* species infection samples. The samples were collected in Faladje. The collected samples were immediately conveyed to the laboratory and processed for the *ex vivo* assay to assess the susceptibility of *P. malariae* and *P. falciparum* parasites to the panel of antimalarial drugs. Briefly, after collecting the samples from volunteers, blood smears were first made, then stained with Giemsa, and the RDT diagnosis was confirmed by microscopy. PCR assays were performed subsequently to confirm the identified *Plasmodium* species and allow the segregation of *P. malariae* mono-infection. Only assays that were performed on mono-infection samples were considered in the data analysis.

## Laboratory procedures

### PCR assay

Blood samples were spotted and blotted on filter paper. Genomic DNA was extracted from dried blood spots using the QIAamp DNA Mini Kit only (Qiagen, Valencia, CA, USA) following the manufacturer's instructions. Real-time qPCR (RT-qPCR) assay was then conducted to identify the different *Plasmodium* species using the small subunit ribosomal RNA 18s (SSU-rRNA) gene as a target for the amplification (21). The RT-qPCR assay was based on a recently developed PCR protocol that targets the SSU-rRNA gene using cooperative primers to increase the sensitivity and specificity of the assay (5).

### Ex vivo drug assay

The antimalarial drugs used for the drug assay were artemether (A9361), quinine (145904), chloroquine (C6628), lumefantrine (L5420), pyrimethamine (P4200000), sulfadoxine (S7821), atovaquone (A7986), piperaquine (C7874), and tafenoquine

(SML0396) (all the compounds were procured from Sigma). All the samples used for the assay were collected within 3–6 hours after the venipuncture. Before starting the drug assay, microscopy was used for the *Plasmodium* species identification and estimation of parasitemia to ensure that samples contained infected red blood cells (RBCs) with at least 80% of asexual ring-stage parasites of *P. malariae* and *P. falciparum*.

The assays were conducted as per the protocol of *P. falciparum* standard culture. Samples were first washed three times with incomplete RPMI-1640 media (10.43 g of RPMI-1640 [Sigma], 5.96 g of HEPES, 2.5 g of $NaHCO_3$, and 2.5 mL of gentamicin [50 mg/mL] for 1 L in $H_2O$).

The washed samples were diluted with packed uninfected RBCs to 0.5% and 1% parasitemia for *P. malariae* and *P. falciparum*, respectively. Then, the diluted samples were then reconstituted to 2% hematocrit using complete RPMI-1640 media and dispensed into the designated wells of 96-well plates. The complete RPMI medium was made of 10.43 g of RPMI-1640, 5.96 g of HEPES, 2.5 g of $NaHCO_3$, 1 mL of hypoxanthine, 5 g of albumax, and 2.5 mL of gentamicin (50 mg/mL) in 1 L of $H_2O$ lacking extra glucose supplement. All the compounds were prepared in dimethyl sulfoxide at 10 mM stock and subsequently diluted in a 1:3 serial dilution with complete media into 10 concentrations, from 10 µM to 0.0005 µM. The assay was set up in duplicates.

The final setup contained infected and uninfected RBCs, diluted drugs, and compounds in a total volume of 120 µL in each well and was maintained in an atmosphere of 2% $O_2$, 5% $CO_2$ and balanced with nitrogen gas. The experiments were stopped after an incubation of 72 hours for *P. malariae* and 48 hours for *P. falciparum*. A volume of 80 µL of the supernatant of each well after the end of the incubation period was removed and replaced with SYBR Green I (Invitrogen, USA) stain in lysis buffer (20 mM Tris [pH 7.5], 5 mm EDTA, 0.008% [wt/vol] saponin, and 0.08% [vol/vol] Triton X-100) at 1× final concentration. Then, the plates were re-incubated in the dark for about 30 minutes, and the fluorescence from each well was determined using the Varioskan Lux multimode microplate fluorescent plate reader (ThermoFisher Scientific, USA) at an excitation of 485 nm, emission of 520 nM and with a gain of 100. About 15 different isolates of each species were tested against each antimalarial compound. Only mono-infections of each of the Plasmodium species confirmed by PCR were considered in the data analysis.

## Data analysis

Fluorescence data were normalized to untreated controls. For each isolate, the assay quality was assessed by computing the $Z'$ factor using positive controls (eight drug-free wells) and negative controls (eight parasite-free red blood cell control wells).

Dose-response curves were plotted with the GraphPad Prism software (version 9) by fitting the data to a curve using a variable slope function and estimating the 50% inhibitory concentrations ($IC_{50}$). $Z'$ values over 0.5 were considered good assays, but each curve is examined by eye for suitability. Some assays with $Z'$ below 0.5 were considered valid depending on factors such as the standard error of the fitted curve and the $IC_{50}$ value.

Statistical tests were conducted using the Mann-Whitney $U$ nonparametric test by GraphPad Prism software version 9. A $P$ value less than 0.05 was considered significant.

## RESULTS

### Evaluation of *ex vivo* efficacy of current antimalarial drugs in inhibiting *P. malariae* and *P. falciparum* growth

This study used *ex vivo* assays to evaluate the susceptibility of *P. malariae* fresh isolates against approved antimalarial drugs (chloroquine, quinine, artemether, pyrimethamine, sulfadoxine, lumefantrine, atovaquone, and tafenoquine) in Ghana, plus piperaquine in Mali (Fig. 2–4; Tables S1 and S2). A total of 11 *P. malariae* isolates and 10 *P. falciparum* were tested and validated for the data analysis, and 12–16 and 11 or 12 isolates were used, respectively, for *P. malariae* and *P. falciparum* assay from Mali.

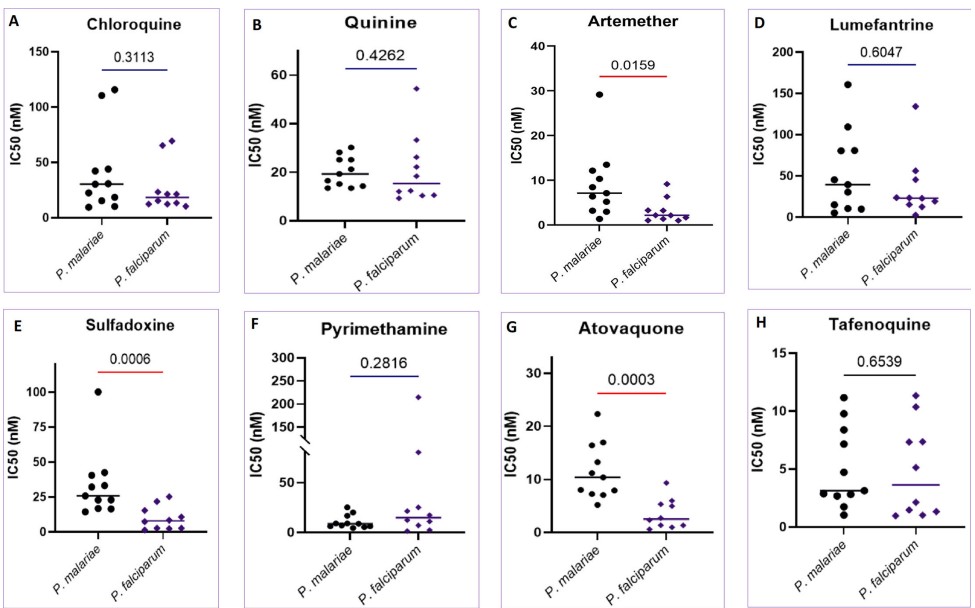

**FIG 2** *P. malariae* and *P. falciparum* comparative susceptibility to current antimalarials from isolates in Ghana. The comparative susceptibility (50% inhibitory concentration [$IC_{50}$, nM]) of the *P. malariae* and *P. falciparum* Ghanaian clinical isolates to conventional antimalarial drugs (A) chloroquine, (B) quinine, (C) artemether, (D) lumefantrine, (E) sulfadoxine, (F) pyrimethamine, (G) atovaquone, and (H) tafenoquine. The number of samples $n = 11$ for *P. malariae* and $n = 10$ for P. *falciparum*. P values have been indicated between the two parasite species (red line = significant difference, violet = not significant difference).

Artemether showed relatively less potent activity against *P. malariae* than *P. falciparum* isolates from Ghana (Tables S1 and S2). The difference was statistically significant in the isolates from Ghana (median $IC_{50}$ equal to 7.12 nM and 2.15 nM, for *P. malariae* and *P. falciparum*, respectively, $P = 0.0159$). However, the isolates from Mali, both species displayed similar susceptibility (median $IC_{50}$ equal to 2.55 nM and 2.05 nM, for *P. malariae* and *P. falciparum*, respectively, $P = 0.4433$) (see Tables S1 and S2; Fig. 2C and 3C). As for lumefantrine, the partner drug in AL combination, the differences were not statistically significant in the isolates from Ghana (median $IC_{50}$ equal to 39.46 nM and 23.06 nM, for *P. malariae* and *P. falciparum*, respectively, $P = 0.6047$), nor in Mali (median $IC_{50}$ equal to 18.34 nM and 18.46 nM, for *P. malariae* and *P. falciparum*, respectively, $P = 0.3873$). However, in Mali, isolates showed a decreased susceptibility to artemether and lumefantrine against *P. malariae* with $IC_{50}$ values above 500 nM as reported previously in Mali (6).

The response of quinine against *P. malariae* was comparable to that of *P. falciparum* in isolates from Ghana, but the *P. malariae* isolates from Mali showed a significantly decreased response to quinine compared to the *P. falciparum* isolates (Fig. 2B and 3B; Tables S1 and S2). Atovaquone and tafenoquine showed less potency against *P. malariae* than *P. falciparum* in Ghanaian isolates and those from Mali (Fig. 2G and H 3G and H; Tables S1 and S2). Chloroquine showed comparable inhibition of *P. malariae* (median $IC_{50}$ = 30.32 nM) and *P. falciparum* (median $IC_{50}$ = 18.32 nM) against the isolates from Ghana. However, in Mali, though similar median $IC_{50}$ values of chloroquine were recorded in *P. malariae* (median $IC_{50}$ = 17.83 nM) and *P. falciparum* isolates (median $IC_{50}$ = 24.05 nM), some isolates had $IC_{50}$s above 300 nM for both species (Fig. 3; Table S2).

Sulfadoxine displayed higher inhibitory potency against *P. malariae* in Ghana (median $IC_{50}$ equal to 25.72 nM and 7.86 nM for *P. malariae* and *P. falciparum*, respectively, $P = 0.0006$). However, in Mali, the sulfadoxine was significantly more potent against *P. malariae* than *P. falciparum* (median $IC_{50}$ equal to 6.62 nM and 21.5 nM for *P. malariae* and *P. falciparum*, respectively, $P < 0.0001$). On the other side, the efficacy of pyrimethamine against *P. malariae* and *P. falciparum* was comparable (Fig. 2Eand 2F 3E and 3F; Table S1

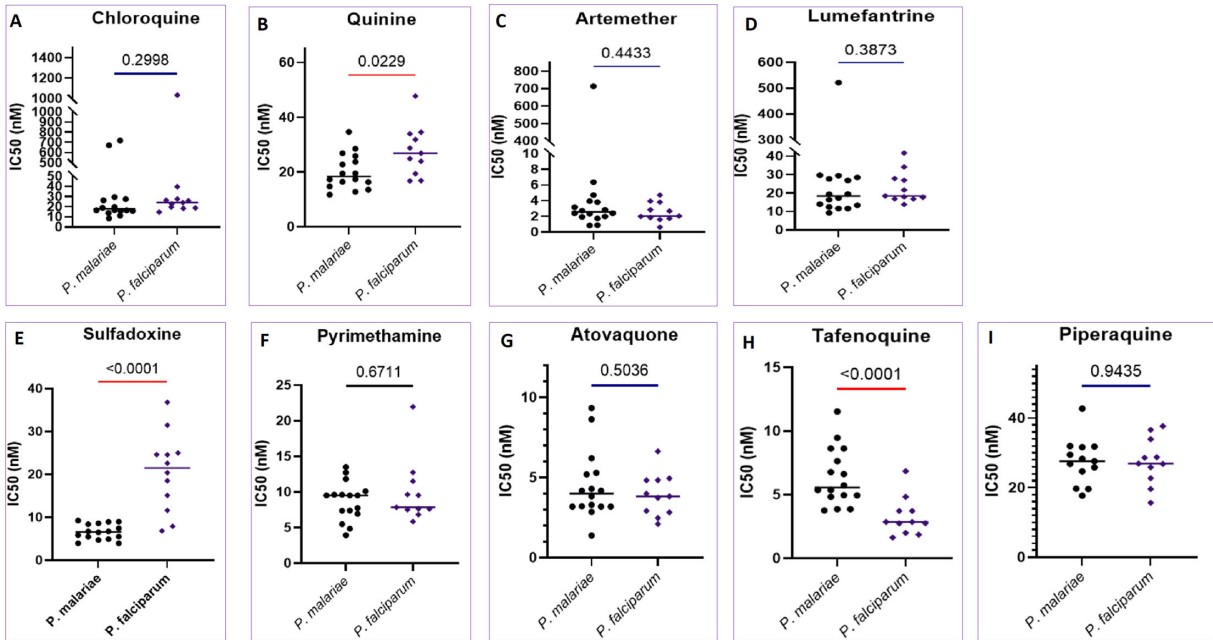

**FIG 3** *P. malariae* and *P. falciparum* comparative susceptibility to current antimalarials from isolates in Mali. The drug compounds activity (IC50 (nM)) to the conventional antimalarial drugs (A) chloroquine, (B) quinine, (C) artemether, (D) lumefantrine, (E) sulfadoxine, (F) pyrimethamine, (G) atovaquone, (H) tafenoquine, and (I) piperaquine against *P. malariae* and *P. falciparum* clinical isolates from Mali. The number of samples n=11 for *P. malariae* and 10 for *P. falciparum*. P-values have been indicated between the two parasite species (red line= significant difference, violet= not significant difference).

and S2). Finally, in Mali, the efficacy of piperaquine against the two *Plasmodium* species was comparable.

## *P. malariae* is more susceptible to some lead discovery candidate antimalarial drugs than *P. falciparum*

The second part of the study tested the efficacy of lead discovery candidate antimalaria drug compounds against *P. malariae* and *P. falciparum* isolates from Mali. The INE963 compound, the most advanced among the candidate drugs tested, showed comparable efficacy against *P. malariae* (median $IC_{50}$ = 1.86 nM) and *P. falciparum* (median $IC_{50}$ = 2.50 nM) isolates (Fig. 4A; Table S3). However, the other candidate drugs were more potent against *P. malariae* than *P. falciparum* (Fig. 4B, C, and D; Table S3), and the difference was statistically significant ($P < 0.05$). Thus, *P. malariae* appeared to be more susceptible to the lead discovery candidate antimalarial compounds than *P. falciparum* (Fig. 4B, C, and D; Table S3).

## DISCUSSION

Malaria control and case management require the use of antimalarial drugs. The existing antimalarials and other malaria-based interventions were largely developed targeting *P. falciparum* biology and its clinical features. The challenge of misdiagnosis and lack of infrastructure to accurately detect the different *Plasmodium* species (22–24) are partly associated with treatment failure and increasing cases of drug resistance (9, 11, 17). In malaria-endemic regions, the non-falciparum species mostly occur as co-infections with *P. falciparum* or exist in lower parasitemia, which leads to benign disease conditions that contrast with many recent reports (5–7, 25). Additionally, *P. malariae* has a 72 hour life cycle, which contrasts with the 48 hour life cycle of *P. falciparum*. With the need to accelerate malaria elimination and eradication agenda, there is an urgent need to evaluate the efficacy of both current and lead candidate antimalarials against *P. malariae* and *P. ovale* infection-driven malaria. To address this, we used *ex vivo* assays to assess

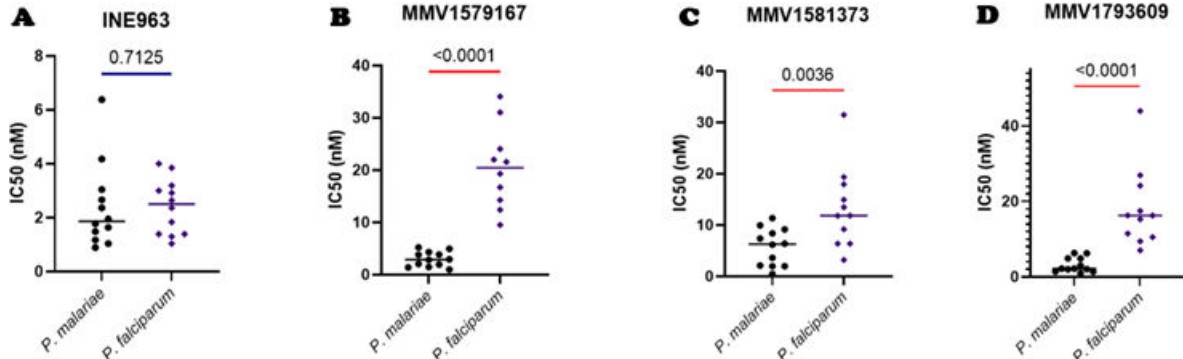

**FIG 4** Susceptibility *P. malariae* and *P. falciparum* to candidate antimalarials drug from isolates in Mali. The different lead compounds were tested for their activity against *P. malariae* (n=12) and *P. falciparum* (n=11) with the IC50 values (nM) plotted against the parasite species for (A) INE963, (B) MMV1579167, (C) MMV1581373, and (D) MMV1793609. The p-values between *P. malariae* and *P. falciparum* isolates have been indicated above the line (red line= significant difference, blue= not significant difference).

the efficacy of existing drugs and new advanced lead discovery compounds against *P. malariae* and *P. falciparum* isolates from Ghana and Mali.

From the data in this study, the potency of artemether against the isolates from Ghana revealed a possible reduced susceptibility of *P. malariae* to this compound compared to *P. falciparum*. However, the $IC_{50}$ values observed from this study were within the range of data previously reported collected on the *ex vivo* drug efficacy of artemether against *P. falciparum* inhibition (26). In addition, there was no statistically significant difference in the responses of *P. malariae* and *P. falciparum* to lumefantrine, the partner drug of artemether in AL combination. This drug combination is one of the main ACTs used in the countries to treat uncomplicated malaria. Notwithstanding this observation, a previous study reported the persistence of positive cases of *P. malariae* after treatment with ACT in Ghana (17). Therefore, there is a need to consider the possible contribution of *P. malariae* infection to ACT resistance monitoring or decreased susceptibility to ACT when examining and treating malaria infections in endemic regions. On the other hand, in Mali, the high $IC_{50}$s observed in artemether and lumefantrine for *P. malariae* susceptibility could imply a decrease in the efficacy of the combination of these two drugs for treating malaria. In a previous study, higher $IC_{50}$s of artemether and lumefantrine against *P. malariae* were recorded in the same area (7). Clinical investigations also revealed the persistence of *P. malariae* cases after treatment with AL (18), suggesting the need to monitor the rate of clearance of the non-falciparum parasite species in patients either in mono-infections or mixed infections.

Quinine inhibited the growth of *P. malariae* isolates more than *P. falciparum* in Mali and Ghana. These results justify the use of quinine in severe case management caused by *P. malariae*, especially because all the $IC_{50}$s are much less than 800 nM, a critical threshold for the resistance of *P. falciparum* to quinine (27).

The efficacy of chloroquine in the growth inhibition of *P. malariae* and *P. falciparum* was comparable in each site. In addition, in Ghana, the $IC_{50}$s were within an acceptable range, indicating reversal to chloroquine-susceptible *P. falciparum* isolates, as reported in other studies. This is an interesting result, since chloroquine is one alternative to ACT for *P. malariae* malaria treatment, according to WHO recommendations. However, the $IC_{50}$s of 7- to 10-fold the usual reference $IC_{50}$ of 100 nM, observed for *P. malariae* and *P. falciparum* in Mali, require more investigation among the different isolates to understand the different mechanisms of chloroquine tolerance in *P. malariae*.

Tafenoquine, an 8-amino-quinoline often used in the radical cure of *P. vivax* malaria or in malaria chemoprevention (28), could be a good alternative for malaria elimination, especially when combined with another drug like the fast-acting artemisinin derivatives (29, 30). Tafenoquine showed more potency in *P. malariae* than *P. falciparum* in Malian isolates, giving hope that the drug could be an effective cure against *P. malariae*

infections. On the opposite, atovaquone exhibited less potency against *P. malariae* in Ghana. This molecule is a hydroxynaphthoquinone often used as a slow-acting drug in combination with proguanil, a biguanide derivative, to treat acute malaria or as chemoprophylaxis, especially in areas with high drug resistance (31). In Mali, no difference was observed between the two species, suggesting this combination could be a good alternative to cure *P. malariae* infections (32).

INE963, a promising candidate antimalarial already undergoing the first clinical trials (Novartis Pharmaceuticals, 2023), demonstrated comparable and efficient growth inhibition efficacy of *P. malariae* and *P. falciparum*. The compound was described as a fast-acting drug as artemisinin (19). While more information is being collected about the safety and potency of the compound in humans, the results of this study suggest that the compound is likely to be effective in curing *P. malariae* cases, given that the range of $IC_{50}$s estimated corroborates existing data (19).

MMV1579167, also known as ELQ-331, is a derivative of ELQ-300 (20) which is a mitochondrial electron transport inhibitor and has been proven to be effective against *P. falciparum* isolates at a range of 4.5–28.2 nM of $IC_{50}$s (33). MMV1579167 has a better bioavailability, is active against liver stages, and is believed to be a good lead candidate drug for long-acting injectable chemoprevention of malaria (33). The results of this study demonstrate the suitability of this drug to cover chemoprevention of *P. malariae* infections, since the $IC_{50}$s were lower than *P. falciparum* values which were within an acceptable range as compared to published data (33).

The data of this study showed a higher growth inhibitory potency of MMV1581373 and MMV1793609 against *P. malariae*, compared to *P. falciparum*. MMV1581373 is a compound at the candidate drug profiling stage, and MMV1793609 is at the preclinical development stage. Even though there is not much information on their efficacy against *P. falciparum*, these compounds showed the possibility of being good candidates for use against *P. malariae* infections.

## Conclusion

Malaria elimination remains challenging, despite all the strategies developed. From case management to prevention, drug efficacy is an essential requirement. In sub-Saharan Africa, aside from *P. falciparum*, involved in the whole process of drug development and surveillance of emergence and spread of drug resistance, *P. malariae* appears to be one of the non-falciparum species that requires more attention, regarding the increasing detection of specific cases and its importance for the achievement of the ultimate malaria elimination. The results of this study showed the strong inhibitory potency of current antimalarial drugs against *P. malariae*. Overall, in Ghana, *P. malariae* displayed acceptable susceptibility as compared to *P. falciparum*, while in Mali, *P. malariae* was proven to have a variable susceptibility to the current antimalarials. However, the data demonstrated the importance of including non-falciparum species in monitoring antimalarial efficacy. In addition, *P. malariae* was more susceptible to most of the candidate drugs tested in Mali, giving the hope of promising prospective integrated control strategies.

## ACKNOWLEDGMENTS

We wish to thank Professor Kelly Chibale (University of Cape Town and H3D Foundation) for providing us all guidance through the project implementation. We express our sincere appreciation to the various health facilities and patients for their participation in the study in Ghana and Mali. We are grateful to Medicines for Malaria Venture for providing compounds to support this research work. We thank Melanie Rouillier for coordinating the compound shipments from MMV. Sincere thanks to all the members of the Molecular Genetics and Cell Biology Research Group at WACCBIP for the insightful scientific discussions on the topic.

This project is funded by the Grand Challenge Africa drug discovery seed grant, ref. GCA/Round10/DD-030 from Science for Africa Foundation. A.S. was supported by WACCBIP-World Bank ACE PhD fellowships, F.A. was supported by the National Institute for Health Research (NIHR) Global Health Research Program 16/136/33, using aid from the UK Government while Y.A. was supported by a Crick African Network.

Conceived and designed the experiments: Y.A., L.D., L.N.A.-E., G.A.A., and A.A.D. Performed the experiments: A.S., F.S., F.A., O.D., J.D.C., F.O.M., M.M., and H.A.D. Analyzed the data: Y.A., L.D., A.S., J.D.C., H.A.D., and M.M. Contributed funding: L.D. and Y.A. Wrote the paper: Y.A., L.D., and A.S. All the authors reviewed the manuscript and approved the final version.

## AUTHOR AFFILIATIONS

[1]West African Centre for Cell Biology of Infectious Pathogens (WACCBIP), College of Basic and Applied Sciences, University of Ghana, Legon, Ghana
[2]Faculty of Pharmacy, Université des Sciences, des Techniques et des Technologies de Bamako (USTTB), Malaria Research and Training Centre (MRTC), Bamako, Mali
[3]Medicines for Malaria Venture (MMV), Geneva, Switzerland

## AUTHOR ORCIDs

Laurent Dembele http://orcid.org/0000-0001-9087-8439
Yaw Aniweh http://orcid.org/0000-0002-8415-2727

## AUTHOR CONTRIBUTIONS

Alamissa Soulama, Data curation, Formal analysis, Investigation, Methodology, Writing – original draft, Writing – review and editing | Fanta Sogore, Data curation, Formal analysis, Investigation, Methodology, Writing – original draft, Writing – review and editing | Felix Ansah, Data curation, Formal analysis, Investigation, Writing – original draft, Writing – review and editing | Ousmaila Diakite, Formal analysis, Investigation, Methodology, Writing – review and editing | Jersley D. Chirawurah, Data curation, Formal analysis, Methodology, Writing – review and editing | Mohamed Maiga, Data curation, Formal analysis, Writing – review and editing | Harry A. Danwonno, Formal analysis, Investigation, Writing – review and editing | Brice Campo, Methodology, Resources, Writing – review and editing | Abdoulaye A. Djimde, Conceptualization, Funding acquisition, Resources, Supervision, Writing – review and editing | Gordon A. Awandare, Conceptualization, Funding acquisition, Supervision, Writing – review and editing | Lucas N. Amenga-Etego, Conceptualization, Methodology, Resources, Supervision, Writing – review and editing | Laurent Dembele, Conceptualization, Data curation, Formal analysis, Funding acquisition, Investigation, Methodology, Supervision, Writing – original draft, Writing – review and editing | Yaw Aniweh, Conceptualization, Data curation, Formal analysis, Funding acquisition, Investigation, Methodology, Project administration, Supervision, Validation, Writing – original draft, Writing – review and editing.

## ETHICS APPROVAL

The current study protocol was reviewed and approved by the Ghana Health Service Ethical Review Committee of Ghana with the reference GHS-ERC:005/12/17 and by the ethical committee of the Faculties of Medicine-Odonto-Stomatology and Pharmacy, University of Science, Techniques and Technologies of Bamako, Mali, with the references numbers N° 2020/296/CE/FMPOS/FAPH, N°2021/70/CE/FMPOS/FAPH, N°2022/03/CE/USTTB, and N°2023/03/CE/USTTB. Only participants or their parent/guardian who provided written informed consent, plus children able to understand the study and who gave their consent, were enrolled in this study. All patients with malaria who consented to participate in the study were enrolled and treated using recommended artemether-lumefantrine (AL) to clear the parasites.

## ADDITIONAL FILES

The following material is available online.

### Supplemental Material

**Supplemental tables (Spectrum02176-24-S0001.docx).** Tables S1 to S3.

### Open Peer Review

**PEER REVIEW HISTORY (review-history.pdf).** An accounting of the reviewer comments and feedback.

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
