## [Reviewer comments · Microbiology Spectrum]

Microbiology Spectrum

Differential *ex vivo* susceptibility of *Plasmodium malariae* and *Plasmodium falciparum* clinical isolates from Ghana and Mali to current and lead discovery candidate antimalarial drugs

Alamissa Soulama, Fanta Sogore, Felix Ansah, Ousmaila Diakite, Jersley Chirawurah, Fatoumata Maiga, Mohamed Maiga, Harry Danwonno, Brice Campo, Abdoulaye Djimde, Gordon Awandare, Lucas Amenga-Etego, Laurent Dembele, and Yaw Aniweh

Corresponding Author(s): Yaw Aniweh, University of Ghana College of Basic and Applied Sciences

Review Timeline:

Submission Date:	August 30, 2024
Editorial Decision:	November 18, 2024
Revision Received:	January 9, 2025
Accepted:	February 9, 2025

Editor: Anna Moniuszko-Malinowska

Reviewer(s): Disclosure of reviewer identity is with reference to reviewer comments included in decision letter(s). The following individuals involved in review of your submission have agreed to reveal their identity: Diogo R. M. Moreira (Reviewer #2)

Transaction Report:

DOI: <https://doi.org/10.1128/spectrum.02176-24>

Re: Spectrum02176-24 (Differential *ex vivo* susceptibility of Ghanaian and Malian *Plasmodium malariae* and *Plasmodium falciparum* clinical isolates to current and lead discovery candidate antimalarial drugs)

Dear Dr. Yaw Aniweh:

Thank you for the privilege of reviewing your work. Below you will find my comments, instructions from the Spectrum editorial office, and the reviewer comments.

Both reviewers recognize the significance of the study on the *ex vivo* susceptibility of *Plasmodium malariae* and *Plasmodium falciparum* isolates to antimalarial drugs. Reviewer #1 commended the work's relevance to malaria control but suggested revisions to improve clarity and reproducibility, including updating the title, addressing contradictions in the data presentation, and providing missing methodological details. Reviewer #2 found the manuscript promising but emphasized the need for better data resolution, clarity in statistical methods, and enhanced discussion around tafenoquine potency and other antimalarials. Both reviewers identified several typographical and formatting errors that need attention.

Recommendations for Authors

- Revise the title to reflect a broader context while maintaining accuracy about the origin of isolates.
- Address contradictions in results (e.g., artemether susceptibility) and provide missing details for piperazine assays and other drug methodologies.
- Consolidate redundant figures and tables to streamline data presentation.
- Enhance discussions of drug potencies and methodologies, ensuring alignment with established literature.
- Correct typographical errors and ensure all scientific names are in italics.
- Provide additional methodological details, including catalogue numbers for drugs and statistical analyses applied.

These revisions will improve clarity, reproducibility, and the manuscript's contribution to the field, moving it closer to publication.

Revision Guidelines

The ASM Journals program strives for constant improvement in our submission and publication process. Please tell us how we

can improve your experience by taking this quick Author Survey.

Sincerely,
Anna Moniuszko-Malinowska
Editor
Microbiology Spectrum

Reviewer #1 (Comments for the Author):

Title: Differential ex vivo susceptibility of Ghanaian and Malian Plasmodium malariae and Plasmodium falciparum clinical isolates to current and lead discovery candidate antimalarial drugs."

Introduction.

This study addresses one of the health challenges facing malaria control interventions: the persistence and drug susceptibility profiles of clinical isolates of Plasmodium malariae and Plasmodium falciparum from Ghana and Mali. Although occurring at a lower frequency than Plasmodium falciparum, Plasmodium malariae infections are often found in many malaria-endemic countries and thus contribute to malaria-associated morbidity and mortality in these countries. The authors also have provided essential hints and comparable analyses of parasites from clinical samples and their responses to standard drugs. However, a few clarifications need to be made, as outlined in the comments section.

Comments to authors

1. The title portrays the "citizenship of the parasites," that is, "Ghanaian" and "Malian" Plasmodium malariae and Plasmodium falciparum. While the samples were isolated from infected patients from Ghana and Mali, it does not necessarily mean that the "exact" parasite genotypes are not found, for instance, in the neighboring countries. The authors should consider revising the title. Possible suggestions for consideration are listed below.
 - i) Differential ex vivo susceptibility of Plasmodium malariae and Plasmodium falciparum clinical isolates from Ghana and Mali to current antimalarials and lead discovery candidate drugs."
 - ii) Ex vivo susceptibility differences of clinical isolates of Plasmodium malariae and Plasmodium falciparum from Ghana and Mali to current antimalarials and lead discovery candidate drugs
2. The authors investigated a panel of drugs used to treat or prevent malaria infection in Africa; however, amodiaquine, a partner drug to artesunate, seemed to have been omitted without any explanation. The authors should provide a comment and justify why this was omitted.
3. The authors in lines 84 - 86 and 150 - 152 have listed a panel of drugs investigated and elaborated on their results. However, the methods of evaluating piperazine are not listed anywhere in the section. How was this piperazine assay evaluated and analyzed? The authors proceeded to illustrate the dataset for piperazine in Figure 3I, and scanty but not very convincing elaboration in lines 241 - 242. Also, why was piperazine not investigated for isolates from Ghana? the authors have stated that "all drugs were procured from Sigma" without including the specific catalog numbers for each drug. These pieces of information are critical and should be included for reproducibility.
4. In lines 217- 219, the authors stated that in isolates from Mali, both species (Plasmodium malariae and Plasmodium falciparum) displayed similar susceptibility to Artemether. However, in the successive statements -lines 223 -225, they stated that Mali isolates showed a decreased susceptibility to artemether and lumefantrine against Plasmodium malariae with IC50 values of 500nM. This is confusing and contradictory. Were the drugs combined, and if so, how? Also, how was this analysis done?
5. Table 1 and Figure 2 represent the same datasets (line 219)-similarly, Table 2 and Figure 3. Table 3 and Figure 4. It is unnecessary to repeat this dataset using different illustrations. The authors should include either the figure or the table in the main manuscript, and the other should be part of the supplemental files.
6. The authors selectively used Mali samples to evaluate the susceptibility of Plasmodium malariae and Plasmodium falciparum against lead discovery candidate antimalarial drugs. Is there an explanation for the choice of these samples? This should be included in the methods.
7. Figures 2D and 2H show a very high variation for lumefantrine and tefanoquine against P. malariae. This variation is not present for isolates from Mali. Can the authors comment on this?
8. All scientific names should be in italics.
9. Authors should check the manuscript for typographical errors, such as the legend for Figure 4, among other sections.

Reviewer #2 (Comments for the Author):

Malaria elimination remains a major challenge. Surveillance of parasite susceptibility for antimalarial drugs, especially in the setting of Sub-Saharan Africa and for P. malariae, is important. Here, Aniweh and co-authors have addressed a quite interesting work on the Surveillance of parasite susceptibility for antimalarial drugs, including reference drugs used in routine, as well as

new potential antimalarial. This manuscript can be of potential interest to the microbiology community. Prior consideration for publication, I would like to encourage the authors to consider the following comments:

A) On the discussion section, authors have stated that: [On the opposite, atovaquone exhibited less potency against *P. malariae* in Ghana. This molecule is an hydroxynaphthoquinone often used as fast-acting drug in combination with proguanil, a biguanide derivative, to treat acute malaria or as chemoprophylaxis, especially in areas with high drug resistance [31].] Authors are recommended to revise it, as Atovaquone is indeed a slowly-acting antimalarial drug.

B) Tables 1-3: Authors are strongly encouraged to specify which range is employed? Is it CI95 %? Please specify.

C) Table 1: it is welcomed to include the abbreviation of the antimalarial drugs in the footnote.

D) For comparison of IC50 values, it was not clear whether authors attempted to apply "Grubbs (Alpha = 0.05)" method to identify outliers. This should be specified.

E) It is recommended to revise the resolution of Figure 1, specify (or replace) the name presumptive, replace the name *P. ovale* (it was a mistake)?

F) Authors are strongly encouraged to discuss in the discussion section whether the potency for tafenoquine is in line with other precedent literature.

Title: Differential *ex vivo* susceptibility of Ghanaian and Malian *Plasmodium malariae* and *Plasmodium falciparum* clinical isolates to current and lead discovery candidate antimalarial drugs."

Introduction.

This study addresses one of the health challenges facing malaria control interventions: the persistence and drug susceptibility profiles of clinical isolates of *Plasmodium malariae* and *Plasmodium falciparum* from Ghana and Mali. Although occurring at a lower frequency than *Plasmodium falciparum*, *Plasmodium malariae* infections are often found in many malaria-endemic countries and thus contribute to malaria-associated morbidity and mortality in these countries. The authors also have provided essential hints and comparable analyses of parasites from clinical samples and their responses to standard drugs. However, a few clarifications need to be made, as outlined in the comments section.

Comments to authors

1. The title portrays the "citizenship of the parasites," that is, "Ghanaian" and "Malian" *Plasmodium malariae* and *Plasmodium falciparum*. While the samples were isolated from infected patients from Ghana and Mali, it does not necessarily mean that the "exact" parasite genotypes are not found, for instance, in the neighboring countries. The authors should consider revising the title. Possible suggestions for consideration are listed below.
 - i) Differential *ex vivo* susceptibility of *Plasmodium malariae* and *Plasmodium falciparum* clinical isolates from Ghana and Mali to current antimalarials and lead discovery candidate drugs."
 - ii) *Ex vivo* susceptibility differences of clinical isolates of *Plasmodium malariae* and *Plasmodium falciparum* from Ghana and Mali to current antimalarials and lead discovery candidate drugs
2. The authors investigated a panel of drugs used to treat or prevent malaria infection in Africa; however, amodiaquine, a partner drug to artesunate, seemed to have been omitted without any explanation. The authors should provide a comment and justify why this was omitted.
3. The authors in lines 84 – 86 and 150 – 152 have listed a panel of drugs investigated and elaborated on their results. However, the methods of evaluating piperazine are not listed anywhere in the section. How was this piperazine assay evaluated and analyzed? The authors proceeded to illustrate the dataset for piperazine in Figure 3I, and scanty but not very convincing elaboration in lines 241 - 242. Also, why was piperazine not investigated for isolates from Ghana? the authors have stated that "all drugs were procured from Sigma" without including the specific catalog numbers for each drug. These pieces of information are critical and should be included for reproducibility.
4. In lines 217- 219, the authors stated that in isolates from Mali, both species (*Plasmodium malariae* and *Plasmodium falciparum*) displayed similar susceptibility to Artemether. However, in the successive statements -lines 223 -225, they stated that Mali isolates showed a decreased susceptibility to artemether and lumefantrine against *Plasmodium malariae* with IC50 values of 500nM. This is confusing and contradictory. Were the drugs combined, and if so, how? Also, how was this analysis done?
5. Table 1 and Figure 2 represent the same datasets (line 219)—similarly, Table 2 and Figure 3. Table 3 and Figure 4. It is unnecessary to repeat this dataset using different illustrations. The authors should include either the figure or the table in the main manuscript, and the other should be part of the supplemental files.
6. The authors selectively used Mali samples to evaluate the susceptibility of *Plasmodium malariae* and *Plasmodium falciparum* against lead discovery candidate antimalarial drugs. Is there an explanation for the choice of these samples? This should be included in the methods.
7. Figures 2D and 2H show a very high variation for lumefantrine and tefanoquine against *P. malariae*. This variation is not present for isolates from Mali. Can the authors comment on this?
8. All scientific names should be in italics.
9. Authors should check the manuscript for typographical errors, such as the legend for Figure 4, among other sections.

RESPONSE TO REVIEWERS

Spectrum02176-24

Reviewer 1

1. The title portrays the "citizenship of the parasites," that is, "Ghanaian" and "Malian" Plasmodium malariae and Plasmodium falciparum. While the samples were isolated from infected patients from Ghana and Mali, it does not necessarily mean that the "exact" parasite genotypes are not found, for instance, in the neighboring countries. The authors should consider revising the title. Possible suggestions for consideration are listed below.
 - i) Differential ex vivo susceptibility of Plasmodium malariae and Plasmodium falciparum clinical isolates from Ghana and Mali to current antimalarials and lead discovery candidate drugs."
 - ii) Ex vivo susceptibility differences of clinical isolates of Plasmodium malariae and Plasmodium falciparum from Ghana and Mali to current antimalarials and lead discovery candidate **drugs**

Response: Thanks for this constructive and improving suggestion. We have now amended the title to read as follow: "Differential ex vivo susceptibility of Plasmodium malariae and Plasmodium falciparum clinical isolates from Ghana and Mali to current and lead discovery candidate antimalarial drugs "

2. The authors investigated a panel of drugs used to treat or prevent malaria infection in Africa; however, amodiaquine, a partner drug to artesunate, seemed to have been omitted without any explanation. The authors should provide a comment and justify why this was omitted.

Response: The current study was part of a bigger project with Medicine for Malaria Venture (MMV) where the selected drug of interest in that study scope did not include amodiaquine. Therefore, it has not been tested. Indeed, we agree that amodiaquine is an important drug to be tested, as it is one of the most used partner drugs to artesunate derivatives in malaria cases management.

3. The authors in lines 84 - 86 and 150 - 152 have listed a panel of drugs investigated and elaborated on their results. However, the methods of evaluating piperazine are not listed anywhere in the section. How was this piperazine assay evaluated and analyzed? The authors proceeded to illustrate the dataset for piperazine in Figure 3I, and scanty but not very convincing elaboration in lines 241 - 242. Also, why was piperazine not investigated for isolates from Ghana? the authors have stated that "all drugs were procured from Sigma" without including the specific catalog numbers for each drug. These pieces of information are critical and should be included for reproducibility.

Response: There is a mistake, piperazine is missing in the list of drugs tested. The sentence at lines 84-85 and 150-150 would be: "The antimalarials used in this study include artemether, lumefantrine, quinine, sulfadoxine, pyrimethamine, atovaquone, chloroquine, tafenoquine, and piperazine." However, the Malian PI of the study decided to investigate piperazine in Mali while this compound was not part of the main study panel as Ghana PI only focused on those main study drugs panel that included only artemether, lumefantrine, quinine, sulfadoxine, pyrimethamine, atovaquone, chloroquine, tafenoquine."

4. In lines 217- 219, the authors stated that in isolates from Mali, both species (*Plasmodium malariae* and *Plasmodium falciparum*) displayed similar susceptibility to Artemether. However, in the successive statements -lines 223 -225, they stated that Mali isolates showed a decreased susceptibility to artemether and lumefantrine against *Plasmodium malariae* with IC50 values of 500nM. This is confusing and contradictory. Were the drugs combined, and if so, how? Also, how was this analysis done?

*Response: Thank you for pointing this out. We do not think there is contradiction. Indeed, in Mali, similar susceptibility to Artemether was observed for both species. However, the statement in lines 223-225 is a comparison with previous work done in the same country where cases of decreased Artemether susceptibility were detected that further lead to our clinical study assessing DHA/Piperazine efficacy in Patients with *P. malariae* as in the clinic we observed some DHA/Artemisinin treatment failure in *P. malariae* infection.*

5. Table 1 and Figure 2 represent the same datasets (line 219)-similarly, Table 2 and Figure 3. Table 3 and Figure 4. It is unnecessary to repeat this dataset using different illustrations. The authors should include either the figure or the table in the main manuscript, and the other should be part of the supplemental files.

Response: many readers like to see the number summarized into a table than just having a figure of a plotted representation. We can consider the figures and then in the core article and tables 1, 2, and 3, will be shifted to supplementary data. Tables 1, 2, and 3 are now Supplementary table 1, 2, and 3.

6. The authors selectively used Mali samples to evaluate the susceptibility of *Plasmodium malariae* and *Plasmodium falciparum* against lead discovery candidate antimalarial drugs. Is there an explanation for the choice of these samples? This should be included in the methods.

Response: In the main study scope and MTA developed with MMV to test the advanced lead discovery compounds, Mali site was selected to implement that objective. In this regards those compounds were only sent to the Malian PI.

7. Figures 2D and 2H show a very high variation for lumefantrine and tafenoquine against *P. malariae*. This variation is not present for isolates from Mali. Can the authors comment on this?

Response: We believe that means that they were some clinical isolates which were less susceptible to lumefantrine in Ghana given that field parasites are diverse in their response to treatments. Clinical isolates have been shown to be mostly having multiple clones. The beauty of this work is to be able to see the diverse response from clinical isolates. This

presents many questions to address going forward. As for tafenoquine, the variation between the site was similar, as part of the development of this standard procedure, we used 7G8 strain to cross compare and validate the site's technical procedures.

8. All scientific names should be in italics

Response: this has been corrected in the full document

9. Authors should check the manuscript for typographical errors, such as the legend for Figure 4, among other sections

Response: this has been completed in the full document

Reviewer 2

A) On the discussion section, authors have stated that: [On the opposite, atovaquone exhibited less potency against *P. malariae* in Ghana. This molecule is an hydroxynaphthoquinone often used as fast-acting drug in combination with proguanil, a biguanide derivative, to treat acute malaria or as chemoprophylaxis, especially in areas with high drug resistance [31].] Authors are recommended to revise it, as Atovaquone is indeed a slowly-acting antimalarial drug.

Response: This is correct; therefore, it has been now revised as suggested to slow-acting.

B) Tables 1-3: Authors are strongly encouraged to specify which range is employed? Is it CI95 %? Please specify.

Response: the CI95 range is the one employed. Since the Tables have been used to plot the figures, we have moved them as supplementary in the manuscript.

C) Table 1: it is welcomed to include the abbreviation of the antimalarial drugs in the footnote

Response: thank for that suggest: here it is please: Footnote: CQ= Chloroquine, QN= Quinine, ART= Artemether. Pyr= Pyrimethamine, SFX= Sulfadoxine, LUM= Lumefantrine, AVQ= Atovaquone, TFN= Tafenoquine, PPQ= Piperaquine

D) For comparison of IC50 values, it was not clear whether authors attempted to apply "Grubbs (Alpha = 0.05)" method to identify outliers. This should be specified

Response: We used Mann-Whitney-U nonparametric test.

The null hypothesis is that “for randomly selected values of IC50s X and Y , from the two populations (two given drug assay group of IC50s), the probability of X being greater than Y is equal to the probability of Y being greater than X “(p-value less than 0.05).

E) It is recommended to revise the resolution of Figure 1, specify (or replace) the name presumptive, replace the name *P. ovale* (it was a mistake)?

Response: this has now been improved.

F) Authors are strongly encouraged to discuss in the discussion section whether the potency for tafenoquine is in line with other precedent literature.

*Response: This has now been discussed as follow: Tafenoquine, an 8-amino-quinoline often used in the radical cure of *P. vivax* malaria or in malaria chemoprevention [28], could be a good alternative for malaria elimination, especially when combined with another drug like the fast-acting artemisinin derivatives [29, 30]. Tafenoquine showed more potency in *P. falciparum* than *P. malariae* in Malian isolates, giving the hope that the drug could be an effective cure against *P. falciparum* infections. Indeed, previous in-vitro testing in *P. falciparum* isolates showed higher EC50 for tafenoquine (DOI: <https://doi.org/10.4269/ajtmh.2002.67.39>). This could probably due to lower susceptibility of malaria parasites to the tafenoquine, in the area, at time of the study.*

Re: Spectrum02176-24R1 (Differential *ex vivo* susceptibility of Ghanaian and Malian *Plasmodium malariae* and *Plasmodium falciparum* clinical isolates to current and lead discovery candidate antimalarial drugs)

Dear Dr. Yaw Aniweh:

Your manuscript has been accepted, and I am forwarding it to the ASM production staff for publication. Your paper will first be checked to make sure all elements meet the technical requirements. ASM staff will contact you if anything needs to be revised before copyediting and production can begin. Otherwise, you will be notified when your proofs are ready to be viewed.

Sincerely,
Anna Moniuszko-Malinowska
Editor
Microbiology Spectrum

Reviewer #1 (Comments for the Author):

I am satisfied with the authors responses and editions made in the revised manuscript.

Reviewer #2 (Comments for the Author):

Authors have responded to my comments.